# Effect of Humus on the Solidification and Stabilization of Heavy Metal Contaminated River Sediment

**DOI:** 10.3390/ijerph20064882

**Published:** 2023-03-10

**Authors:** Huimin Gao, Hong Tao, Yang Yang, Qingyang Che, Qinyi Tang, Yong Gu

**Affiliations:** School of Environment and Architecture, University of Shanghai for Science and Technology, Shanghai 200093, China; 202541981@st.usst.edu.cn (H.G.);

**Keywords:** river sediment, solidification and stabilization, organic matter, heavy metal, speciation

## Abstract

To better reutilize heavy metal contaminated river sediment containing organic matter, the sediments in a river located in Chongming District, Shanghai were collected and Portland cement was used as a curing agent along with commercial organic matter to conduct the solidification/stabilization experiment. The unconfined compressive strength and heavy metal leaching concentrations of solidified blocks with different water content, organic matter content, and cement content were tested and analyzed to determine the optimal ratio. The effects of fulvic acid (FA), humic acid (HA), and an HA/FA ratio on the solidification and stabilization, as well as the speciation of heavy metals in sediment before and after solidification and stabilization, were studied. The results showed that when the organic content of the sediment is 6.16%, the water content is 65% and the cement content is greater than 38%, so the curing effect proves to be satisfactory. Fulvic acid has a stronger inhibiting effect on cement hydration than humic acid, and its consumption in the curing process is more significant. The addition of humic acid contributes to the stabilization of heavy metals, while the increase in fulvic acid greatly weakens the stability of heavy metals. The exchangeable state of heavy metals in the sediment has been reduced to varying degrees after solidification and stabilization. The research results can provide a basis for the reclamation and utilization of heavy metal contaminated river sediment with organic matter.

## 1. Introduction

With the rapid industrial development in China, water contamination has become increasingly prominent. As sediment is an important part of the aquatic ecosystem, the contaminated sediments in rivers act as the convergence site of different pollutants and the source of secondary pollution, thereby attracting wide concern. Malodorous black or eutrophic river sediments usually come with numerous pollutants, such as organic matter and heavy metals. Under certain conditions, these pollutants would be released into water and cause secondary pollution through dissolution, ion exchange, desorption, and other means [1]. Cement is commonly used as a curing agent in the solidification and stabilization of heavy metal contaminated sediments due to its high efficiency, low cost, and quick effect [2].

However, the type and content of the organic matter in the sediment have an important impact on the solidification effect. The composition of the organic matter in the soil is very complex and the molecular structure is also very different. However, on the whole, it can be divided into two categories: animal and plant residues that have not been decomposed or fully decomposed, and humus resulting from complete decomposition or degradation [3]. The International Humus Society (IHSS) and the Chinese soil science community divide humus into three categories according to solubility: humic acid (HA), fulvic acid (FA), and humin. As humin is inert in soil, most current studies on humic substances focus on humic acid and fulvic acid [4]. Zhang et al. [5] established the mathematical model of various factors of humic acid components in sludge stabilized soil and the strength of cement stabilized soil, showing that both humic and fulvic acid have a weakening effect on cement reinforcement from the perspective of microstructure mechanisms, as well as physical and chemical properties. Yang [6] and Shao [7] also illustrated in their studies that organic matter would inhibit the hydration reaction in solidified soil and weaken the cementation of hydration products and sediment particles, thus loosening the structure and hindering the formation of solidified soil and further reducing its strength. There are two stages for organic matter to affect stabilization strength. In the first stage, the compressive strength decreases along with the increase in the organic matter, with the two showing an almost linear relationship. In the second stage, when organic matter content is higher than a certain value, 0 c (the limit effect content of organic matter on strength), no apparent decrease in the strength is noted with the continuous increase in organic matter. Through batch field experiments, D. J. Ashworth et al. [8] found that there is a strong positive correlation between the solubility of Cu, Ni, Zn, and organic matter. Cui [9] concluded that the solid solution of different heavy metal ions in cement clinker minerals is selective, with most of them taking the form of vacancy filling or replacement. By adding humic acid and fulvic acid to heavy metal contaminated sediment, Li et al. [10] argued that the addition of fulvic acid could activate heavy metals in the sediment, dramatically increasing their soluble form, while the addition of humic acid significantly reduces the amount of soluble heavy metals, creating a passivation effect.

Previous studies have so far discussed the effect of organic matter content on compressive strength after solidification and on the stabilization of heavy metals, where organic matter is generally removed as pollutants to repair the sediment/soil. However, the complexation of organic matter with heavy metals and its inhibition of hydration reaction has not been systematically studied. Organic matter has a significant effect on the adsorption and release of heavy metals [11] and is susceptible to complexation with heavy metals, thus reducing the dissolution of heavy metals. Moreover, organic matter causes the soil to have a greater water capacity and plasticity, greater expansion, and low permeability, and makes the soil acidic, all of which impede the hydration reaction, which means that organic matter diminishes the stabilizing effect of the curing agent on heavy metal contamination. Therefore, organic matter exerts an interactive rather than a singular influence on the stability of heavy metal contaminated soil after solidification and stabilization. In this study, Portland cement-425 was adopted for the solidification of the heavy metal contaminated sediment to explore what the best solidification conditions were and how organic matter content, humic acid, fulvic acid, and HA/FA ratio influence solidification and stabilization. Through excitation-emission matrix fluorescence (EEM), the speciation changes of humus before and after solidification were analyzed, and the effects of humic substances on the compressive strength of the solidified block in curing, as well as on heavy metal leaching concentration and speciation, were studied. The interaction mechanism of organic matter on the solidification and stabilization of heavy metal contaminated soil is further investigated.

## 2. Materials and Methods

### 2.1. Experimental Materials

(1)Test sediment: the test sediment was taken from a river channel in Chongming District, Shanghai. The sediments with high organic matter and low organic matter were collected in two places for the subsequent experiment to adjust the content of the organic matter. The sampling depth was 0~20 cm. The physical and chemical properties of the sediment are shown in Table 1. The additional amount of heavy metal salts Cd (NO_3_) _2_⋯4H_2_O, Cu (NO_3_) _2_⋯3H_2_O, Pb (NO_3_) _2_ is based on the Soil Environmental Quality Standard for Soil Pollution Risk Control of Agricultural Land (GB15618-2018), simulating heavy metal contaminated sediment. Composite heavy metal contaminated sediment was prepared after even stirring and aging for 1 month, with the moisture content kept at about 30%. The heavy metal content before and after sediment pollution is shown in Table 2.(2)Curing agent: Shanghai Conch PO.42.5 ordinary Portland cement was selected.(3)Commercial HA/FA was purchased from Shanghai Sinopharm Chemical Reagents Co., Ltd.

### 2.2. Experimental Method

The influence of various factors on the unconfined compressive strength and toxicity leaching of the solidified test block of sediment were investigated in single-factor experiments by setting different cement content (14%, 18%, 22%, 26%, 30%, 34%, 38%, 42%, 46%), organic matter content (4.58%, 5.84%, 6.16%, 6.68%, 7.73%, 8.78%, 9.31%, 9.62%, 10.88%) and moisture content (50%, 55%, 60%, 65%, 70%, 75%, 80%, 85%) of the solidified body. The content of organic matter in the sediment is adjusted by controlling the mixing amount of high organic matter sediment and low organic matter sediment. The curing effect is based on the compressive strength of 7 days, 14 days, and 28 days (subsequent representations are 7 d, 14 d, and 28 d) as reference indicators. The leaching concentration of Cd, Cu, and Pb in the solidified body is determined by the toxicity leaching test. During the experiment, the original sample of the sediment added with heavy metals is mixed evenly. After aging for one month, the required sediment, Portland cement and water are weighed according to the design dosage, and the subsequent experiment is carried out after mixing, molding, demolding, and curing in the curing box until the design age.

Based on the optimal curing conditions and proportions, the test blocks are prepared by adding different amounts of commercial humic acid and fulvic acid into the sediment, and the effects of humic acid and fulvic acid on the curing stability are explored with 28 d compressive strength and metal leaching concentration as indicators.

At the same time, Tessier’s five-step continuous extraction procedure is used to analyze the chemical speciation forms of heavy metals in the experimental sediment and the sediment solidification and stabilization test block [13], so as to explore the effects of solidification on the distribution of heavy metal forms in the test block.

Separation of humic acid and fulvic acid: the soil samples before and after solidification are pretreated and mixed with 0.1 mol/L NaOH in the ratio of solid to liquid 1:15, and the basic filtrate A (humic acid + fulvic acid) is obtained by shaking culture, centrifugation, and filtration; 1.5 mol/LHNO3 is added to filtrate A until the pH < 2.0, and acidic filtrate B (fulvic acid) is obtained through filtration. The content is determined by the total organic carbon/total nitrogen analyzer, and the organic matter before and after curing is characterized by three-dimensional fluorescence. Combined with heavy metal leaching and speciation analysis data, the influence of organic matter on the solidification and stabilization is explored.

### 2.3. Testing Method

With reference to the Standard for Soil Test Methods (GB/T 50123-1999) [14], the sediments were made into standard test blocks with a diameter of 50 mm and a height of 50 mm, and the standard test blocks that had completed the curing at the specified age were tested for unconfined compressive strength using DYE-300S full-automatic cement compression and bending tester. In accordance with the Solid Waste Leaching Toxicity Leaching Method-Acetic Acid Buffer Solution Method (HJ/T300-2007) [15], and with the employment of extractant 2 # (pH = 2.65 ± 0.05), the concentration of heavy metal leaching solution was compared with the provisions of Class V limit values in the Environmental Quality Standard for Surface Water (GB3838) [16]. The concentration of heavy metal in the leaching solution was detected by Inductively Coupled Plasma Mass Spectrometer (ICP-MS, NexION 300X, Perkin-Elmer, Waltham, MA, USA). The organic matter content was determined by the total organic carbon analyzer, and the three-dimensional fluorescence analysis was performed with an F-7000 fluorescence spectrometer (Hitachi, Tokyo, Japan). Three-dimensional fluorescence analysis was determined using an F-7000 fluorescence spectrometer (Hitachi, Tokyo, Japan). The scanning procedure of the fluorescence spectrometer was as follows: the TOC concentration of the sample to be measured was diluted to about 5 mg/L, Mill-Q ultrapure water was used as the blank, the instrument PMT voltage was set to 700 v, the excitation wavelength (Ex) range was 200–500 nm with a step of 5 nm, the emission wavelength (Em) range was 300–600 nm with a step of 2 nm, the sample scanning speed was 12,000 nm/min, and the response time was 0.1 s. The measurements were subtracted from the Mill-Q ultrapure water 3D fluorescence data to eliminate the effect of water Raman scattering, while the Rayleigh scattering was set to 0.

### 2.4. Statistical Analysis

The sample data calculation and graphs were completed using Microsoft Excel 2016 and Origin 2019. Pearson correlation and significant difference analysis were conducted by IBM SPSS 22.0.

## 3. Results and Discussion

### 3.1. Factors Influencing the Cement Solidification and Stabilization of Heavy Metal Polluted Sediment in River Course

Unconfined compressive strength (UCS) is an important mechanical property index of cement curing materials. High compressive strength requirements are put forward for the materials used in engineering applications such as subgrade landfill, river slope protection, and bank protection. This experiment takes the USC cured for 7 d, 14 d, and 28 d as the index to explore the influence of various factors on the curing effect.

Figure 1 shows the change curve of the compressive strength of the solidified block under different conditions. It can be seen from Figure 1a that the USC of the solidified block shows an upward trend with the increase in the cement content and curing age. As the cement content increases from 14% to 46%, the UCS of the solidified block rises from 0.65 MPa to 3.25 MPa at 28 d. When the cement content is higher than 30%, the strength of the test block at 7 d is 1.3 MPa, which reaches the strength of the cement-stabilized material for the subbase layer of roads of Class II and below, as specified in the Technical Rules for Construction of Highway Pavement Base Course (JTGTF20-2015) issued by the Ministry of Transport. When the cement content is greater than or equal to 38%, its 28-d strength is greater than or equal to 2.3 MPa, which meets the required strength of ISER II solidified soil, as specified in the Technical Standard for Construction of Ecological Slope Protection and Bank Protection by In Situ Use of Dredged Mud (DG/TG 08-2331-2020) [17]. When the cement content is less than 15%, the curing effect cannot be clearly seen, and the compressive strength of the test block is lower than the detection limit of the instrument.

Figure 1b shows the relationship between USC and organic matter content in the sediment. There is a highly significant correlation between the organic matter content and UCS of curing blocks (7 d: *p* < 0.01, r = −0.863; 14 d: *p* < 0.01, r = −0.896; 28 d: *p* < 0.01, r = −0.940). As the curing age increases, the strength grows slightly. It can be concluded that organic matter has a certain inhibitory effect on the hydration reaction of solidified blocks. It can be seen from Figure 1b that the impact of organic matter on UCS reaches the limit when the content of the organic matter reaches about 10%, that is, the compressive strength decreases with the increase in the content of organic matter within the limit value. As is shown, with the content of organic matter increasing from 5% to 10%, the compressive strength decreases from 3 MPa to 1.8 Mpa, but when the content of the organic matter exceeds the limit value, the compressive strength almost does not change. The influence of organic matter on the compressive strength is mainly due to the following two reasons: (1) the fulvic acid and humic acid in organic matter make the hydration reaction environment acidic, which is not conducive for the formation of gel materials during the reaction [18]; (2) the adsorption between fulvic acid and cement minerals leads to the formation of an adsorption layer on the mineral surface to delay the hydration reaction. The decomposition of fulvic acid on hydration products, such as calcium sulphoaluminate hydrate, calcium aluminate hydrate, and calcium ferric aluminate hydrate crystals, has destroyed the formation of the cement-soil structure, showing the characteristics of chemical weathering [19].

The water content has an extremely significant impact on the UCS of the solidified block (7 d: *p* < 0.01, r = −0.926; 14 d: *p* < 0.01, r = −0.905; 28 d: *p* < 0.01, r = −0.931). From Figure 1c, it can be seen that the compressive strength of the solidified block decreases as a whole as the water content increases from 50% to 65%, and the 28-d compressive strength of the solidified block decreases rapidly from 2.5 MPa to 1.4 MPa. When the water content exceeds 65%, the decline trend of strength slows down. The compressive strength is provided by the combination of crystal skeleton and gel, as well as the combination of gel and unhydrated cement particles. The existence of pores has a negative impact on the strength [20]. With the increase in water content, the porosity of the system increases. At the same time, the pH value of the reaction environment decreases because the soluble organic matter (such as fulvic acid) in the sediment is dissolved in the water, which hinders the hydration reaction, thus reducing the compressive strength with the increase in water content. Saride et al. [21] show that the pH of the solidified soil with high HA content decreases with the increase in age. The hydration reaction is unstable under acidic conditions and the hydration products calcium silicate hydrate and calcium silicate hydrate are dissolved, resulting in the reduction in UCS.

Taking the leaching concentration of heavy metals as the index, the effect of cement content, organic matter content, and water content was analyzed. Figure 2a shows the effect of water content on the leaching concentration of heavy metals. As shown in the figure, the leaching concentration of Cu (*p* < 0.05; r = −0.815) and Pb (*p* < 0.05; r = −787) is negatively correlated with the water content, and the Cd leaching concentration curve fluctuates up and down, but generally shows a downward trend. With the increase in water content, some active heavy metals in the system are dissolved in water and lost during the curing process. In addition, with the increase in water content, the soluble organic matter in the sediment combines with the heavy metals to form a stable complex, resulting in the decrease in the leaching concentration. When the water content is 65% and above 75%, the content of Cd, Cu, and Pb in the leaching solution of the solidified soil conforms to the Class III limit value, as specified in the Environmental Quality Standard for Surface Water (GB3838) [16].

The relationship between organic matter content and Cd, Cu, and Pb leaching concentration is shown in Figure 2b. With the increase in organic matter content, the Cu (*p* < 0.01; r = −0.900) and Pb (*p* < 0.01; r = −0.881) leaching concentrations decrease. When the organic matter content exceeds 6.26%, the decline trend slows down. HA and FA have a good affinity for Cu and Pb and form stable inner bound complexes [22] through covalent bonds. The Cd leaching concentration curve fluctuates up and down without an obvious trend. Cadmium is generally considered as the most mobile trace metal in the soil, and its complexing ability with organic matter is weak. When organic matter complexes with Cd, it will compete with Cd for the adsorption point on the surface of soil particles, reducing the adsorption of Cd on the soil and demonstrating a complex interaction [23,24]. Therefore, the change of organic matter content has a disturbance on the Cd leaching concentration, yet results in no obvious change trend. When the content of organic matter is greater than or equal to 6.16%, the content of Cd, Cu, and Pb in the leaching solution of solidified soil conforms to the Class III limit value, as specified in the Environmental Quality Standard for Surface Water (GB3838) [16].

Figure 2c shows the relationship between the leaching concentration of heavy metals and the amount of cement added. With more cement added, the leaching concentration of Cd (*p* < 0.05; r = −0.731), Cu (*p* < 0.05; r = −0.753), and Pb (*p* < 0.01; r = −0.882) shows a downward trend. When the cement content reaches 38%, the decline of the leaching concentration of heavy metals begins to slow down. Higher cement content brings about more intensive solidification reactions. Loose soil particles are cemented into larger soil aggregates to form a more compact soil structure, which helps adsorb and fix heavy metals in the solid phase [25]. However, with excessive cement, the generated gel materials are connected with each other, thus reducing the adsorption of heavy metals and resulting in the slowing down of the decline of heavy metal leaching concentration [26]. When the cement content is higher than 38%, the content of Cd, Cu, and Pb in the leaching solution of the solidified soil conforms to the Class III limit value specified in the Environmental Quality Standard for Surface Water (GB3838) [16].

To sum up, the curing effect proves satisfactory when the organic matter content of the sediment is 6.16%, the water content is 65%, and the cement content is greater than 38%.

### 3.2. Effect of Humus on the Solidification and Stabilization of Heavy Metal Contaminated River Sediment

Humic acid and fulvic acid are the main components of humus in organic soil. Fulvic acid has a significant effect on curing and stabilization [22]. In the process of humification, the content of fulvic acid gradually decreases and the content of humic acid gradually increases. Generally, the ratio of humic acid to fulvic acid (DI) is called the humification index, which can be used as an indicator of the maturity of organic matter [27]. In order to explore the impact of these two humic acids on the solidification and stabilization during the humification of sediment, this study conducted qualitative and quantitative research on humic acid and fulvic acid based on relatively satisfactory solidification conditions adopted in previous studies.

Figure 3 shows the relationship between the content of humic acid and fulvic acid, the HA/FA ratio (the total content of humic acid and fulvic acid is controlled at 1%), and the UCS of solidified soil. It can be seen from the Figure that when the content of humic acid and fulvic acid increases, respectively, the compressive strength of the solidified soil sharply decreases. When the content of humic acid reaches 4%, the compressive strength of solidified soil decreases to 0, and when the content of fulvic acid reaches 2%, the compressive strength of solidified soil decreases to 0. The reason for this is that humic acid has a strong chemical affinity and selectivity for calcium ion (Ca^2+^) and it reacts with hydrated lime [Ca (OH) _2_], forming insoluble substances adsorbed on the surface of cement and clay particles that disturb the hydration and pozzolanic reaction [28]. At the same time, acid ions will decompose the cement hydration products and the cementation formed by other reactions, destroying the cementation body and the solidified skeleton of solidified silt, thus affecting the compressive strength of the solidified soil [29]. The molecular weight of fulvic acid is small and the acidity is stronger. Compared with Figure 3a,b, fulvic acid has a more significant weakening effect on the compressive strength of solidified soil than humic acid. It can be seen from Figure 3c that the strength of solidified organic soil changes linearly with the increase in the HA/FA ratio, and gradually increases with the increase in the HA/FA ratio. When the HA/FA ratio is less than one, fulvic acid plays a leading role, with low strength of the solidified organic soil and small slope of the reinforcement curve. When the HA/FA ratio is greater than one, the strength of the solidified organic soil is improved and the slope of the strength growth curve is large, indicating that the main material that has an adverse impact on the curing strength is fulvic acid.

Humus has an important impact on the adsorption and release of heavy metals. It can be seen from Figure 4a that the leaching concentration of heavy metals decreases with the increase in humic acid content (Cu: *p* < 0.01, r = −0.871; Pb: *p* < 0.01, r = −0.971; Cd: *p* < 0.01, r = −0.922). Figure 4b reflects that the leaching concentration of heavy metals increases with the increase in fulvic acid content (Cu: *p* < 0.01, r = −0.991; Pb: *p* < 0.01, r = −0.974; Cd: *p* < 0.01, r = −0.902). The humus in the solidified soil has two competing mechanisms for heavy metal elements: on the one hand, due to the binding ability of humus and heavy metal ions, humus combines with metal ions to form coordination compounds; on the other hand, the inhibition of humus on cement hydration reaction leads to the weakening of the stabilizing effect of the hydration products on heavy metals [30]. It is usually easy for humic acid and metal ions to form insoluble chelates [31], and fulvic acid has more carboxyl and alcohol groups than humic acid, which makes fulvic acid more acidic and further leads to stronger inhibition on cement hydration and weaker fixation of hydration products on heavy metals. Figure 4c shows that with the increase in the relative amount of humic acid and the decrease in the relative amount of fulvic acid in the mixed acid, the leaching concentration of Cd, Cu, and Pb decreases and the stability of heavy metals increases, which is similar to the experimental results of the pure humic acid group. This could be attributed to the fact that with the decrease in fulvic acid concentration, the complexing ability of fulvic acid weakens, and the heavy metal-fulvic acid complex that is activated and dissolved into water decreases. With the gradual increase in the humic acid concentration, humic acid combines with more heavy metals. In the alkaline environment of the cement sediment, the heavy metal-humic acid complex is precipitated and fixed on the surface of the sediment particles, so the stability of heavy metals is improved and the leaching concentration gradually decreases. Xiao et al. [32] also found that the addition of humic acid can change the form distribution of heavy metals and improve the stability. In the soil with humic acid, with the increase in humic acid addition, the exchangeable, carbonate Pb and Cd in the soil gradually drops and is significantly lower than that in the control group, while the iron-manganese bound, organic, residual Pb, and Cd in the soil gradually grows and is significantly higher than that in the control group.

Comparing Figure 4a–c shows that, with the same dosage, the leaching concentration of heavy metals in different groups is as follows: fulvic acid group > mixed acid group > humic acid group. The addition of humic acid contributes to the stabilization of heavy metals, while the increase in fulvic acid greatly weakens the stability of heavy metals. Therefore, the present study contends that the stability of heavy metals improves with the development of the humification process. With the high humification degree of the sediment, it is appropriate to adopt solidification and stabilization, chemical stabilization, and other technologies to further improve the stability of heavy metals.

### 3.3. Speciation Changes of Heavy Metals in Sediment before and after Solidification

Previous studies [33,34,35] have found that heavy metals will combine with OH− or silicate and solidify in calcium salt when mixed with cement-based materials, and they can also be adsorbed in C-S-H gel to enter the crystal structure. Pb can usually be adsorbed on the surface of hydration products. Most CuO will participate in hydration to form Cu-Ca-Si minerals, and Cd will replace calcium ions in the calcium oxide layer and interlayer structure of C-S-H gel. The speciation of heavy metals in soil or sediment can be divided into exchangeable (F1), carbonate-bound (F2), iron-manganese oxidized (F3), organic-bound (F4) and residual (F5), according to Tessier’s five-step extraction method. From Figure 5, it can be seen that Cd in the experimental sediment before solidification mainly exists in the form of F1 and F2, with contents of 0.39 mg/kg and 0.17 mg/kg, respectively, accounting for 50.65% in F1 and 22.08% in F2. Cu mainly exists in the form of F2 and F4, with the content of F2 standing at 92.75 mg/kg, accounting for 50.81%, and the content of F4 standing at 40.18 mg/kg, accounting for 22.01%. Pb mainly exists in the form of F2 and F3. Heavy metal leaching is closely related to the distribution of speciation [36], especially the unstable F1, F2, and F3. F1 is mobile and can be easily absorbed by organisms, featuring direct biological toxicity. It can be seen in the figure that the experimental sediment before solidification has a high potential risk to the environment. After curing, the F1 content of Cd decreases, while the F2 and F3 content increases. The F2 content of Cu and Pb decreases and the F3 content increases. It can be seen that the exchangeable states of Cd, Cu, and Pb are reduced to varying degrees after solidification, which indicates that solidification reduces the leaching ability of Cd/Cu/Pb, which is of great significance for the reduction in heavy metal toxicity.

### 3.4. Characterization of Organic Matter in Sediment before and after Solidification

Table 3 shows the content changes of humic acid and fulvic acid in the sediment before and after solidification. It can be seen that the content of organic matter in the sediment decreases during the solidification process. On the one hand, humic acid precipitates after complexing with bivalent metal cations in the sediment. On the other hand, in the hydration process, Ca^2+^ is a weak cationic bond bridge, which can form the outer ring complex of the calcium bond complex with humic acid [37].

The three-dimensional fluorescence spectrum of humus extracted from the sediment before and after solidification is shown in Figure 6. The three-dimensional fluorescence spectrum can help obtain the fluorescence intensity information when the emission wavelength and excitation wavelength change simultaneously, so as to infer the configuration change of humus [38].

Based on the observation and analysis of Figure 6, the number and position of fluorescence peaks in the four samples of humus extracted before and after solidification are basically the same, indicating that they contain basically the same fluorescence groups and that no new fluorescence groups are generated during the solidification process.

The fluorescence characteristic index, FI, is the ratio of the fluorescence intensity at the excitation wavelength of 370 nm and the emission wavelengths of 450 nm and 500 nm, which can characterize the source of humus in fluorescent soluble organic matter. When FI > 1.9, it indicates that the fluorescent dissolved organic matter mainly comes from the microbial metabolic activity of the water body [39]. In this study, the FI of the sample is 1.96~4.61, suggesting that the humus extracted from the sample mainly comes from microbial metabolism, mainly from autogenesis.

The fluorescence peak that characterizes the four samples, peak A, is located at Ex/Em = 225–260 nm/270–320 nm and is a complexine-like acid fluorescence peak that characterizes protein-like substances. One study [40] classifies the protein-like fluorescence peak into the tryptophan-like fluorescence peaks(Ex/Em = 270–290 nm/320–350 nm) and the complexine-like acid fluorescence peak (Ex/Em = 270–290 nm/300–320 nm), according to the length of the emission wavelength. According to the fluorescence intensity analysis of the peak A of different samples, the fluorescence intensity of PC (B) is significantly lower than that of RS (B), and the extraction solution of B is fulvic acid extracted from the sediment, which indicates that the consumption of fulvic acid is large during the solidification process. Fulvic acid combines with aluminum on the surface of mineral particles. When the complex formed reaches a certain concentration, it will precipitate out of the solution [41]. This is consistent with the previous conclusion that fulvic acid has a more conspicuous weakening effect on curing strength.

Peak B at Ex/Em = 250–260 nm/450–470 nm is a fulvic acid-like fluorescence peak that characterizes humic substances.

## 4. Conclusions

(1)With the increase in the cement content and curing age, the UCS of the solidified block increases, and the leaching concentration of heavy metals decreases. Organic matter has an inhibitory effect on the hydration reaction of the solidified block. When the content of the organic matter in the sediment increases within the limit value, the UCS of the solidified block shows a linear downward trend, the leaching concentration of Cu and Pb decreases, and the leaching concentration of Cd does not change significantly. Water content is negatively correlated with the UCS and the heavy metal leaching concentration of the solidified block. When the water content exceeds 60%, the decline of UCS slows down.(2)The weakening effect of commercial humic acid/fulvic acid on the strength of cured samples is more obvious than that of natural organic matter, no matter if it is mixed acid or a single humic acid/fulvic acid, both of which have consistent inhibiting and delaying effects on the hydration reaction of cement. Among them, fulvic acid has a stronger weakening effect on the strength of cured samples than humic acid. Humic acid shows a stabilizing effect on heavy metals, while fulvic acid demonstrates an activating effect on heavy metals. With the increase in the humification index (DI), the leaching concentration of heavy metals decreases and their stability improves.(3)The main speciation of heavy metals in raw sediment includes unstable F1, F2, and F3, with high mobility and high biological toxicity. The key to the stabilization of heavy metals is the treatment of unstable heavy metals. Solidification converts unstable heavy metals into stable heavy metals through adsorption, encapsulation, and co-precipitation, effectively reducing the environmental risk of the sediment.(4)The fluorescent groups in the sediment before and after curing are essentially the same, and no new fluorescent groups are generated. The fluorescence characteristic index of the sediment samples before and after solidification is 1.96~4.61, indicating that the humus extracted from the samples mainly comes from microbial metabolism, mainly from autogenous sources. After solidification, the content of organic matter in the sediment decreases and the consumption of fulvic acid is more significant than that of humic acid.

## Figures and Tables

**Figure 1 ijerph-20-04882-f001:**
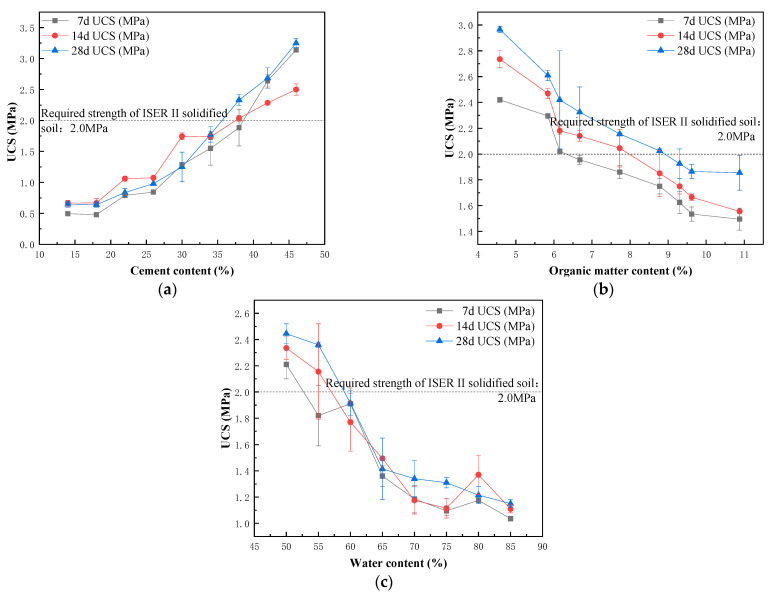
(**a**) Compressive strength of solidified block with different cement contents; (**b**) Compressive strength of solidified block with different organic matter contents; and (**c**) Compressive strength of solidified block with different water contents.

**Figure 2 ijerph-20-04882-f002:**
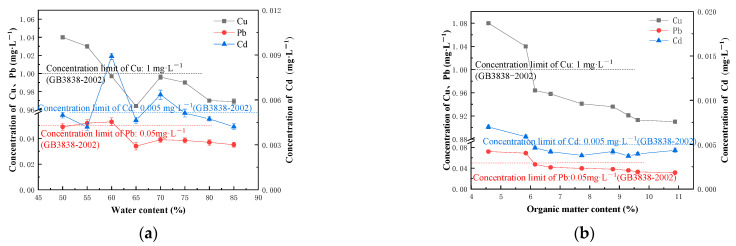
(**a**) Heavy metal leaching concentration with different water content; (**b**) Heavy metal leaching concentration with different organic matter content; and (**c**) Heavy metal leaching concentration with different cement content.

**Figure 3 ijerph-20-04882-f003:**
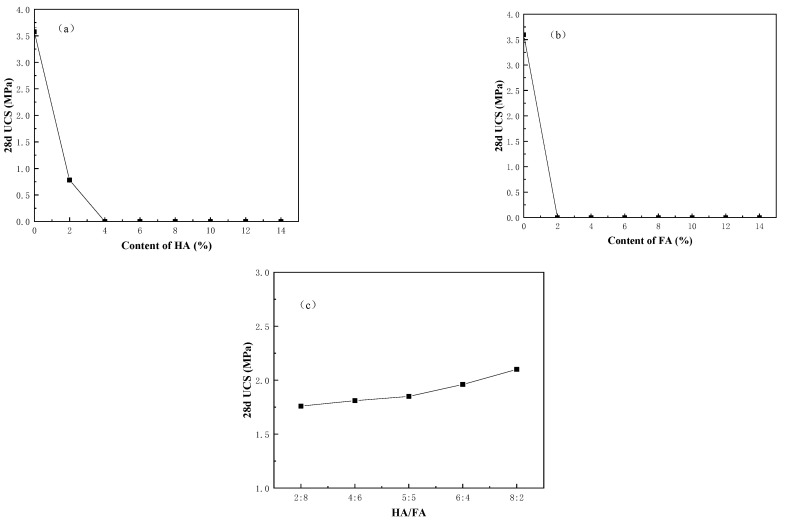
(**a**) Effect of humic acid on UCS; (**b**) Effect of fulvic acid on UCS; and (**c**) Effect of different HA/FA ratios on UCS.

**Figure 4 ijerph-20-04882-f004:**
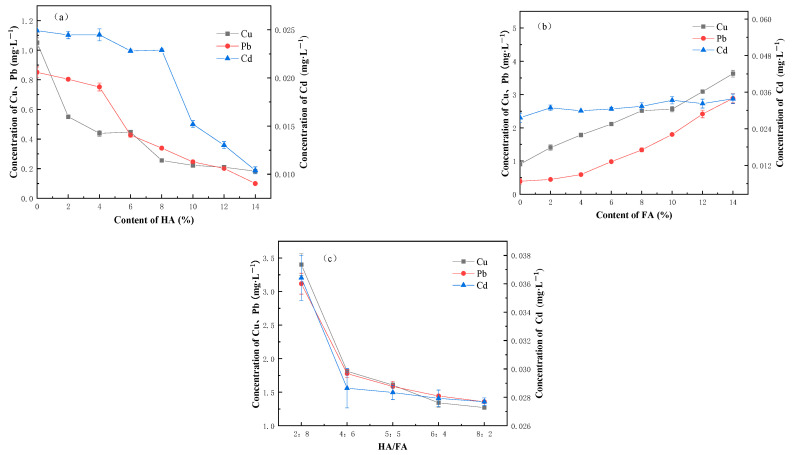
(**a**) Effect of humic acid on heavy metal leaching; (**b**) Effect of fulvic acid on heavy metal leaching; and (**c**) Effect of HA/FA ratio on heavy metal leaching.

**Figure 5 ijerph-20-04882-f005:**
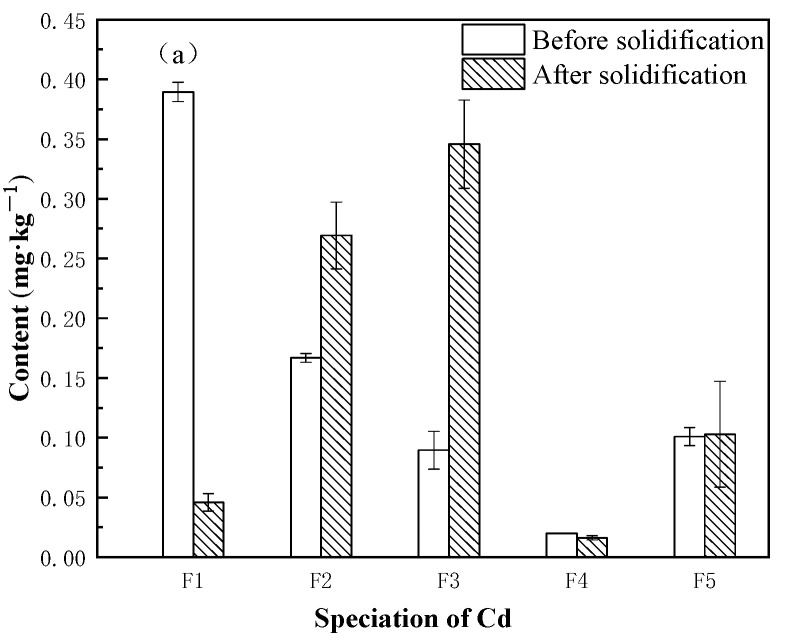
(**a**) Speciation changes of Cd before and after solidification; (**b**) Speciation changes of Cu before and after solidification; and (**c**) Speciation changes of Pb before and after solidification.

**Figure 6 ijerph-20-04882-f006:**
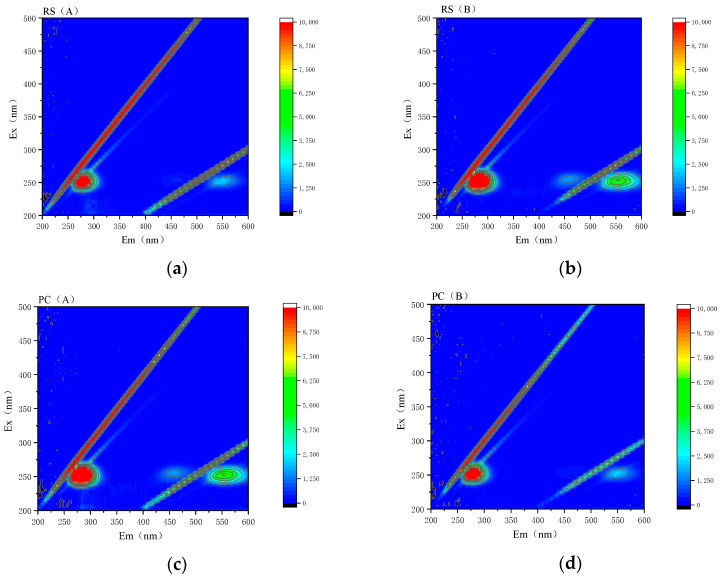
(**a**) Raw sediment A extract; (**b**) Raw sediment B extract; (**c**) Solidified sediment A extract; and (**d**) Solidified sediment B extract.

**Table 1 ijerph-20-04882-t001:** Main Physical and Chemical Properties of Tested Sediment.

Sediment	Classification	Organic Matter Content (%)	Plasticity Index	C (%)	N (%)	H (%)	S (%)
High organic sediment	Organic soil	10.88	14.53	16.16	1.55	2.37	1.12
Low organic sediment	Low liquid limit silt	4.58	14.62	2.88	0.20	0.67	0.22

**Table 2 ijerph-20-04882-t002:** Heavy Metal Content of Sediments before and after Contamination (mg·kg^−1^).

Heavy Metal	Cd	Cr	Ni	Pb	Zn	Cu
High organic sediment	0.27	83.06	41.02	32.29	91.61	27.60
High organic sediment (after pollution)	0.95	83.06	41.02	170.22	91.61	151.72
Low organic sediment	0.28	49.53	31.44	17.04	62.80	28.27
Low organic sediment (after pollution)	0.99	49.53	31.44	197.98	62.80	149.86
Background value	0.14	70.2	29.9	25.5	86.1	27.2
GB15618filter value	0.3	200	100	120	250	100

Note: The background values are based on the heavy metal background values of sediments in Shanghai [12].

**Table 3 ijerph-20-04882-t003:** Content of humic acid and fulvic acid in solidified sediment.

Group	Humic Acid Content (mg/g)	Fulvic Acid Content (mg/g)
RS	1.89	1.83
PC	1.74	1.76

Note: RS and PC refer to raw sediment and Portland cement solidified sediment, respectively.

## Data Availability

The datasets generated and analyzed during the current study are available from the corresponding author on reasonable request.

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
