# Peer review of "Effect of Humus on the Solidification and Stabilization of Heavy Metal Contaminated River Sediment"

_ijerph, 2023, doi:10.3390/ijerph20064882_

Round 1
Reviewer 1 Report
Comments
In this manuscript presented by Gao et al., the effects of humus on the solidification and stabilization of heavy metal-contaminated river sediment were investigated.
Overall, from the perspectives of environmental science and health, this study has its merit since it is dealing with heavy metals, and developing method for decontaminations. However, there are several concerns need to be urgently addressed by the authors before further considerations for publication.
The materials and methods used for all the experiments were not clearly explained in detail. For example, where were those sediment samples exactly collected and how were the sample sites selected? How many samples in total? I guess it is two according to Table 1. If there were any replicates, which it should have, Table 1 should provide those property parameters as mean with standard deviation etc.
Table 2 is another major concern. Between the high- and low organic sediments, the metals (Cd, Pb, and Cu) were adjusted to different values, especially for Pb. Then it raised concerns that whether it was still appropriate to compare the high and low organic sediment samples, in terms of the effect of organic %. Furthermore, the concentrations of other metals (listed in Table 2) were different between samples, it would also cause the inconsistency for comparison.
As described in the Abstract (line 17), …” the best proportion is as follows…”, however in the method part, the authors failed to provide more details about how to get the best proportion, how many conditions were tested and how to determine the best one? Please provide such related information.
Last but not least, the data presented in this manuscript was not properly analyzed, at least not properly presented, since there was no statistical analysis at all. It would lead to questions like how could we know the effects were significant or even there were any effects.
Author Response
Mar 4, 2023
Dear reviewer,
Thank you very much for giving us the opportunity to revise our manuscript, entitled as “Effect of humus on the solidification and stabilization of heavy metal contaminated river sediment” (ijerph-2254705). We very much appreciate the comments and suggestions from the editor and reviewers, which are valuable for improving the quality of our manuscript. The manuscript has been extensively revised according to all the comments. Detailed point-to-point responses are presented as follows.
Thank you again for your attention to our manuscript. It would be highly appreciated if you are kind to give a favorable consideration on it. We look forward to hearing from you soon.
Yours sincerely,
Hong Tao, PhDSchool of Environmental and ArchitectureUniversity of Shanghai for Science and TechnologyShanghai, 200093, ChinaTel: 86-21-55275979; fax: 86-21-55275979E-mail: taohong126126@126.com
Response to Reviewer 1:
The materials and methods used for all the experiments were not clearly explained in detail.
Response: Thanks for your comment. The test substrate was taken from a river in Jianjian Town, Chongming District, Shanghai. The low organic matter substrate (121°27′21.48″, 31°40′35.62″) at the confluence with the external river and the high organic matter substrate (121°27′18.83″, 31°40′25.78″) away from the external river were sampled in two places to facilitate the adjustment of the organic matter content of the substrate, and the sampling depth was A total of two samples were taken from 0 to 20cm.
- Whether different sample metal concentrations affect the comparison of high versus low organics samples.
Response: Thanks for your comment. To facilitate adjustment of the organic matter content of the substrate, high and low organic matter substrates were taken in two locations. The heavy metal salts Cd(NO3)2-4H2O, Cu(NO3)2-3H2O and Pb(NO3)2 were added to the substrate to simulate heavy metal contaminated substrate. The composite heavy metal contaminated substrate was produced by mixing well and aging for one month. During the ageing process, the organic matter affects the migration and transformation behaviour of the heavy metals (Collins, R. N., Merrington, G., McLaughlin, M. J., and Morel, J. L. (2003). Transformation and fixation of Zn in two polluted soils by changes of pH and organic ligands. Aust. J. Soil Res. 41, 905–917.). such that the heavy metal content of the high organic matter substrate differs from that of the low organic matter substrate. The difference in heavy metal content between each group of experimental mud after mixing is less than the difference between the contaminated high organic matter mud and the contaminated low organic matter mud, which can be accommodated by the experimental error.
- This information is crucial for assessing results and calculating metal concentrations for the geological background. There was no data on texture and organic matter (LOI).
Response: Thanks for your comment. I am very sorry, firstly I did not state it properly in the abstract, the correct statement should be that the curing stabilisation was better at an organic matter content of 6.46%, a cement admixture of 35.34% and a moisture content of 55.00% (line 16-18). Secondly, to determine the best conditions, I used the metal leaching concentration and the unconfined compressive strength of the cured body as indicators of organic matter content (set gradients: 4.58%, 5.84%, 6.16%, 6.68%, 7.73%, 8.78%, 9.31%, 9.62%, 10.88%), cement admixture (set gradients: 14%, 18%, 22%, 26%, 30%, 34%, 38%, 42%, 46%), and moisture content (set gradients of 50%, 55%, 60%, 65%, 70%, 75%, 80%, 85%) as variables to investigate the optimum curing stabilization conditions through single factor experiments (line 108-112).
- The data presented in this manuscript was not properly analyzed, as no statistical analysis was available.
Response: Thanks for your comment. Pearson correlation and significant difference analyses with IBM SPSS 22.0 have been added.( lines 162-165 & lines 189-190 & lines 207-208 & lines 232-233 & lines 244-245 & line 260 & lines 322-325)
All changes have been marked in red in the MS.

Reviewer 2 Report
This is a manuscript about the effects of fulvic acid (FA), humic acid (HA) and HA/FA ratio in the composition of humus on solidification and stabilization of soil with cement, as well as the speciation of heavy metals in sediment before and after solidification and stabilization. The addition of humic acid showed contribution to the stabilization of heavy metals, while the increase of fulvic acid greatly weakened the stability of heavy metals. The exchangeable state of heavy metals in the sediment was shown to be reduced to varying degrees after solidification and stabilization.
There are some gaps of information and writing issues that need to be addressed before this manuscript can be considered for publication in IJERPH:
Line 14: maintain past tense ("were" tested)
Line 16: "results showed" (past)
Line 18: "fulvic" acid (not "furic")
Line 34: cross out "causing secondary pollution" because it is repetitive
Line 41: rephrase to "humus resulted from complete decomposition or degradation" (otherwise it sounds like the humus is being decomposed rather than being the result of decomposition process)
Lines 72-74: citation needed for this hypothesis/claim. What "various ways"? Support this statement with background literature available.
Lines 79-84: all verbs must be at past tense because the work has already been performed
Line 90: was (not "is")
Lines 91-92 and throughout the manuscript: Chemical formulae must be written properly, using proper subscripts and superscripts (when ionic)
Line 101: IJERPH has a very diverse readership. Please explain to your readers what this jargon means "7d, 14d and 28d"!
Lines 124-onward: Maintain past tense verbs!
Materials & Methods: very incomplete descriptions of methods. ICP-MS procedure must be described, including instrumentation description with settings used, or a citation provided detailing the procedure. Same for fluorescence spectrometry.
Results and discussion (rather than "analysis")
Line 211: was analyzed
Line 229: complexes
Figure 2 and figure 4: y-axes labels should read "Concentration of..."
Lines 276-277: "2+" must be superscript, "2" subscript
Lines 347-350 & 383 & 390-403: a blank space should be inserted between value and units of measurement reported in these paragraphs
Line 371: "2+" superscript
Line 418: "if" or "whether" is missing before "it"
Line 419: rephrase to "both of which..."
Line 430: replace "basically" with "essentially"
The last reference in the bibliography is written in Cyrillic characters or similar. Please check!
Author Response
Mar 4, 2023
Dear reviewer,
Thank you very much for giving us the opportunity to revise our manuscript, entitled as “Effect of humus on the solidification and stabilization of heavy metal contaminated river sediment” (ijerph-2254705). We very much appreciate the comments and suggestions from the editor and reviewers, which are valuable for improving the quality of our manuscript. The manuscript has been extensively revised according to all the comments. Detailed point-to-point responses are presented as follows.
Thank you again for your attention to our manuscript. It would be highly appreciated if you are kind to give a favorable consideration on it. We look forward to hearing from you soon.
Yours sincerely,
Hong Tao, PhDSchool of Environmental and ArchitectureUniversity of Shanghai for Science and TechnologyShanghai, 200093, ChinaTel: 86-21-55275979; fax: 86-21-55275979E-mail: taohong126126@126.com
Response to Reviewer 2:
Line 14: maintain past tense ("were" tested).
Response: Thanks for your comment. This has been corrected.
- Line 16: "results showed" (past)
Response: Thanks for your comment. This has been corrected.
- Line 18: "fulvic" acid (not "furic").
Response: Thanks for your comment. This has been corrected.
- Line 34: cross out "causing secondary pollution" because it is repetitive.
Response: Thanks for your comment. This has been This has been crossed out (line 35).
- Line 41: rephrase to "humus resulted from complete decomposition or degradation" (otherwise it sounds like the humus is being decomposed rather than being the result of decomposition process)
Response: Thanks for your comment. This has been corrected (line 42).
- Lines 72-74: citation needed for this hypothesis/claim. What "various ways"? Support this statement with background literature available.
Response: Thanks for your comment. Organic matter causes the soil to have a greater water capacity and plasticity, greater expansion and low permeability, and makes the soil acidic, all of which impede the hydration reaction, which means that organic matter diminishes the stabilizing effect of the curing agent on heavy metal contamination.( Lines 73-77)
- Lines 79-84: all verbs must be at past tense because the work has already been performed.
Response: Thanks for your comment. This has been corrected (lines 80-85).
- Line 90: was (not "is").
Response: Thanks for your comment. This has been corrected (line 93).
- Lines 91-92 and throughout the manuscript: Chemical formulae must be written properly, using proper subscripts and superscripts (when ionic)
Response: Thanks for your comments. This has been corrected (lines 94-95).
- Line 101: IJERPH has a very diverse readership. Please explain to your readers what this jargon means "7d, 14d and 28d"!
Response: Thanks for your comment. The curing effect is based on the compressive strength of 7days, 14days and 28days as reference indicators. Explanation has been added to the manuscript (lines 115-116).
- Lines 124-onward: Maintain past tense verbs!
Response: Thanks for your comment. This has been corrected (lines 141-onward).
- Materials & Methods: very incomplete descriptions of methods. ICP-MS procedure must be described, including instrumentation description with settings used, or a citation provided detailing the procedure. Same for fluorescence spectrometry.
Response: Thanks for your comment. A detailed description has been added to the manuscript (lines 149-150 & lines 152-161).
- Results and discussion (rather than "analysis")
Response: Thanks for your comments. This has been corrected (line 166).
- Line 211: was analyzed
Response: Thanks for your comments. This has been corrected (line 231).
- Line 229: complexes
Response: Thanks for your comments. This has been corrected (line 250).
- Figure 2 and figure 4: y-axes labels should read "Concentration of..."
Response: Thanks for your comment. These have been corrected in the figure 2 and figure 4.
- Lines 276-277: "2+" must be superscript, "2" subscript
Response: Thanks for your comment. These have been corrected (lines 299-300).
- Lines 347-350 & 383 & 390-403: a blank space should be inserted between value and units of measurement reported in these paragraphs
Response: Thanks for your comment. These have been corrected (lines 371-374 & 408 & 415-427).
- Line 371: "2+" superscript
Response: Thanks for your comment. This has been corrected (line 396).
- Line 418: "if" or "whether" is missing before "it"
Response: Thanks for your comment. This has been corrected (line 443).
- Line 419: rephrase to "both of which..."
Response: Thanks for your comment. This has been corrected (line 444).
- Line 430: replace "basically" with "essentially"
Response: Thanks for your comment. This has been corrected (line 455).
- The last reference in the bibliography is written in Cyrillic characters or similar. Please check!
Response: Thanks for your comment. This has been corrected (line 547).
All changes have been marked in red in the MS.

Round 2
Reviewer 1 Report
can be accepted.